# Community-level impacts of spatial repellents for control of diseases vectored by *Aedes aegypti* mosquitoes

Quirine A. ten Bosch[1¤a]*, Joseph M. Wagman[2¤b], Fanny Castro-Llanos[3], Nicole L. Achee[1], John P. Grieco[1], T. Alex Perkins[1]*

**1** Department of Biological Sciences and Eck Institute for Global Health, University of Notre Dame, Notre Dame, Indiana, United States of America, **2** Uniformed Services University of the Health Sciences, Bethesda, Maryland, United States of America, **3** United States Naval Medical Research Unit No. 6, Callao, Peru

¤a Current address: Quantitative Veterinary Epidemiology, Wageningen University and Research, Wageningen, the Netherlands
¤b Current address: PATH Center for Malaria Control and Elimination, Washington DC, United States of America
* Quirine.tenbosch@wur.nl (QAtB); taperkins@nd.edu (TAP)

**Data Availability Statement:** Code used to perform the statistical and mathematical analyses is available on GitHub (https://github.com/quirine/

## Abstract

Spatial repellents (SRs) reduce human-mosquito contact by preventing mosquito entrance into human-occupied spaces and interfering with host-seeking and blood-feeding. A new model to synthesize experimental data on the effects of transfluthrin on *Aedes aegypti* explores how SR effects interact to impact the epidemiology of diseases vectored by these mosquitoes. Our results indicate that the greatest impact on force of infection is expected to derive from the chemical's lethal effect but delayed biting and the negative effect this may have on the mosquito population could elicit substantial impact in the absence of lethality. The relative contributions of these effects depend on coverage, chemical dose, and housing density. We also demonstrate that, through an increase in the number of potentially infectious mosquito bites, increased partial blood-feeding and reduced exiting may elicit adverse impacts, which could offset gains achieved by other effects. Our analysis demonstrates how small-scale experimental data can be leveraged to derive expectations of epidemiological impact of SRs deployed at larger scales.

## Author summary

Mosquito control strategies that reduce bites to humans through multiple, non-lethal modes of action may be important in controlling mosquito-borne diseases where insecticidal strategies are ineffective. Assessing how effective such tools are in reducing infections is not clear-cut due to the multiple ways these products affect mosquitoes' behavior and life cycle. We introduce a paired experimental and mathematical framework to analyze and combine data from experiments on the several effects of a transfluthrin formulation and assess its public health impact. We show that, while product-induced lethality accounts for the majority of the product's impact, delayed blood feeding can, through its

SRCommunityEpi) and experimental data are available on Open Science Framework (DOI 10. 17605/OSF.IO/J9CKS).

**Funding:** This work was supported and funded by the Bill & Melinda Gates Foundation (gatesfoundation.org) (Grant #48513): "A push-pull strategy for Aedes aegypti control." QAtB was supported by a graduate student fellowship from the Eck Institute for Global Health (globalhealth.nd. edu) at the University of Notre Dame. JMW was supported by a grant from the Uniformed Services University (usuhs.edu). The funders had no role in study design, data collection and analysis, decision to publish, or preparation of the manuscript.

**Competing interests:** The authors have declared that no competing interests exist.

negative impact on mosquito population sizes, elicit its own substantial impact. Adverse effects of increased partial blood-feeding and reduced exiting could offset gains achieved by other effects such delayed blood feeding and lethality. Our model offers a way of synthesizing the results of feasible experiments at small scales to assess public health impact at large scales.

## Introduction

Recent decades have seen significant progress in reducing the burden of malaria, in part due to the success of insecticide treated nets (ITNs) [1]. Public health successes of vector control have also been achieved for dengue and yellow fever, but none of these successes were sustainable [2]. The continued expansion of dengue virus (DENV) and others transmitted by *Aedes aegypti* [2]—combined with setbacks in the development of effective vaccines for dengue and Zika [3]—underscores the growing need for more effective vector-based strategies for controlling these diseases.

Spatial repellents (SRs) are one of many vector control tools currently being evaluated for their potential public health value. SRs consist of products aimed at reducing contact between humans and mosquitoes by preventing mosquito entrance into human-occupied spaces, interfering with the detection of humans, or disrupting the blood-feeding process [4]. The effectiveness of SR products has been demonstrated in entomological studies against a range of vector species [5, 6] and has also been indicated in epidemiological studies of malaria [7, 8]. Meanwhile, diversion of mosquitoes to untreated houses was found to offset beneficial community-level effects in some studies [9, 10] but this is thought to be outweighed by the beneficial effects in other studies [11, 12].

Spatial repellency constitutes a range of effects distinct from other chemically induced effects, such as contact irritancy and lethality [4]. Most compounds exhibit a combination of these modes of action [13]: *i.e.*, repellency or deterrency (preventing entry), irritancy or expellency (promoting exit), reduced biting, and lethality (Fig 1). In addition, excitatory effects can promote unintended behaviors, such as reduced exiting or increased probing, wherein hyper-agitated mosquitoes remain in treated spaces and attempt multiple, partial blood meals prior to full engorgement and oviposition [14]. At the same time, these behaviors can also lead to delayed oviposition or increased mortality associated with host contact during blood feeding. Similarly, reduced exiting does not necessarily result in an increased probability of blood feeding at a particular house, as it may be accompanied by reduced blood feeding due to sensory overstimulation [6].

How the balance of these (species-specific) effects influences the public health value of SRs is likely to depend on details of the product's formulation; e.g., the active chemical ingredient and its dosage [13]. Furthermore, public health value at the community level may differ substantially across settings with different types of housing structures and densities, outdoor and indoor mosquito predators, and with other ongoing vector control efforts, among other factors. The interplay of these effects complicates projections of the epidemiological impact of SRs and other vector control products that exert multiple effects on mosquito behavior and bionomics.

Examining the many possible combinations of effects that different products have and addressing their context-specificity in epidemiological field trials would be extremely costly, if not altogether unfeasible. Modeling has therefore been used to sharpen our understanding of the epidemiological impact of combinations of effects of other vector control tools for malaria,

## THE MULTIFACETED EFFECTS OF VECTOR CONTROL PRODUCTS AGAINST *AEDES AEGYPTI*

Mosquito life events

Transmission time line

**BASELINE**

EIP   Infectious period

No transmission   Transmission

**REDUCED LIFE SPAN**

μ
φ
†

No transmission

**DELAYED BLOOD FEEDING / PROLONGED GONOTROPHIC CYCLE**

o < 1
α < 1
ρ

> q

No transmission   Transmission

**INCREASED PARTIAL BLOOD FEEDING / PROLONGED GONOTROPHIC CYCLE**

α > 1
o < 1

No transmission   Transmission

Fully blood fed   Partially blood fed

μ = Acute mortality
φ = Delayed mortality
o = Relative gonotrophic cycle length

α = Relative biting rate
q = Exit rate
ρ = Repellency

= Oviposition
† = Mosquito death
= Susceptible
= Infected

= Untreated house

= Treated house

**Fig 1. Multiple effects SR products on mosquito behavioral and bionomic traits and impact on mosquito-borne pathogen transmission.** Each row presents a potential scenario of mosquito life events after it has become infected, in the absence (top row) or presence (other rows) of SRs. On the left, flight behavior in search for blood meals or oviposition sites (containers) is depicted in untreated (gray) and treated houses (pink). On the right, the mosquito's life span after human-to-mosquito transmission is shown, with duration of the extrinsic incubation period (EIP) (dashed green line) and infectious period (solid green line). Blood meals (full or partial) may result in mosquito-to-human transmission (red human) following the EIP, but not before (white humans). Once a mosquito is fully blood-fed, it searches an oviposition site, after which the next gonotrophic cycle starts. Scenarios presented from top to bottom: A) baseline in absence of the SR, B) reduced mosquito life span resulting from lethality effects of the SR ($\mu$ or $\phi$), C) reduced blood feeding and prolonged gonotrophic cycle resulting from repellency ($\rho$), excitatory effects ($\alpha$ and o), and expellency, and D) increased blood feeding and prolonged gonotrophic cycle due to increased partial blood feeding ($\alpha$) but prolonged time until fully blood-fed (1/o).

such as insecticide-treated nets (ITNs) [15, 16] and attractive toxic sugar baits [17]. Insights have been gained on both the individual- and community-level impact of various products and combinations thereof [18, 19], including explorations of circumstances with adverse diversion effects [15], impacts on insecticide resistance [19–21], and product decay [21].

In contrast to the focus of previous modeling work on malaria, we developed a model of SR product impact specifically for diseases vectored by *Ae. aegypti*. Our model considers a broad range of acute and delayed effects, including repellency (decreased entry) and expellency (increased exit), excitatory effects on biting, decoupling biting rates from time until oviposition by accounting for both full and partial blood meals, and examining the context specificity of epidemiological impact. For consistency, the product effect is hereby referred to as 'lethality' (chemical toxicity) and the bionomic effect as 'mortality' (to include both lethality and hazards experienced when moving from one house to another). We informed the model with data from small-scale entomological experiments of transfluthrin on *Ae. aegypti* behavior and bionomics. Because these experiments used differing means of application or delivery of transfluthrin and mosquito populations with different levels of resistance to pyrethroids, this work does not constitute a projection of the epidemiological impact of a singular product formulation. However, these data demonstrate how this model can be used to estimate the relative change in DENV force of infection brought about by a product with such properties introduced into a community at a given coverage.

## Results

### Product effects

By combining data from experimental studies with a set of probabilistic models, we derived model parameters for six distinct entomological effects of the transfluthrin treatment: delayed blood feeding ($\alpha$) expressed as a proportion of the mean rate of blood feeding in the absence of SR, a prolonged gonotrophic cycle (1/$o$), the probability of death upon encountering the SR ($\mu$), increases in mortality rates ($\phi$), the probability of repellency ($\rho$), and the relative change in exit rates ($q_T/q_U$).

**Exposure to transfluthrin resulted in increased time until fully blood-fed and an increased propensity for partial blood feeding.** We performed experiments to estimate the effect of exposure to low ($8.4 \times 10^{-7}$ g/L) and high ($12.6 \times 10^{-7}$ g/L) dosages of transfluthrin on the rate at which mosquitoes took blood meals. We distinguished partial and full blood meals. Contrary to a full blood meal, a single partial blood meal did not lead to full engorgement and thus required further feeding attempts.

We fitted a multinomial model to the feeding data (see Materials and Methods) and found that both full and partial biting rates (a) (Table 1) were affected significantly upon exposure to the experimental dosages of transfluthrin in comparison to estimates from the control (Table 2). The average time until blood feeding was increased by 28% (95% highest density interval HDI: 13–43%) after exposure at low dosage and similarly for high dosage (27%, HDI:

**Table 1. Baseline parameters for force of infection framework.**

| Symbol | Description | Units | Default | Reference |
|---|---|---|---|---|
| $a$ | Biting rate | days$^{-1}$ | 0.76 | [14] |
| $1/o$ | Duration of gonotrophic cycle | days | 4 | [50] |
| $g_U$ | Daily mortality rate (indoors) | days$^{-1}$ | 0.18 | [51] |
| $g_\tau$ | Daily mortality rate (outdoors) | days$^{-1}$ | 0.18 | [51] |
| $q_U^{-1}$ | Time spent in untreated house | days | 2 | |
| $\tau$ | Time spent per transit event | days | 0.3 $q_U^{-1}$ | [52] |
| $n$ | Duration of the extrinsic incubation period | days | 14 | [53] |
| b | Probability of mosquito to human infection | | 0.5 | |
| c | Probability of human to mosquito infection | | 0.5 | |
| X | Human infection prevalence | | 12.5 | |

12–44%) (Fig 2). This result was largely driven by a reduction in the rate at which full blood meals were taken, with the average time until a full blood meal increasing relative to the control by 46% (HDI: 27–65%) (low) and 74% (HDI: 50–100%) (high). Part of this reduction in the rate at which full blood meals were taken was offset by an increase in partial blood meals due to incomplete imbibing and/or aberrant probing, but not enough to result in a net increase in blood-feeding rate. The partial blood-feeding rate increased insignificantly relative to the control in response to low exposure by 5% (HDI: -19-26%) but did increase significantly after high exposure 28% (HDI: 10–44%). The probability of blood feeding over time by dosage was derived using eqns. S3 and S5 (Fig 2). We parameterized the biting rate ($a_C$) to be a fraction α of 0.78 (HDI: 0.69–0.88) (low) and 0.79 (HDI: 0.69–0.89) (high) and the oviposition rate ($o_C$) a fraction 0.69 (HDI: 0.59–0.78) (low) and 0.57 (HDI: 0.50–0.66) (high) (Table 2). The estimated effects are, in line with the Ross-McDonald framework, based on the assumption of exponentially distributed time until blood feeding. Relatively high rates of blood feeding at early time points, particularly in control experiments, indicate a departure from this assumption. This is likely due to starving of mosquitoes prior to the experiments (Fig 2). Estimates of $a_C$ based on parametric survival models with different assumptions about time-varying hazards of blood

**Table 2. Fitted model parameters depicting effects of transfluthrin under experimental laboratory and field conditions.**

| Experiment | Model framework parameter description | Model framework parameter symbol | Fitted equation | Fitted parameters (95% confidence interval: lower bound, higher bound) | Model framework default (HDI) |
|---|---|---|---|---|---|
| Blood feeding | Fraction of baseline blood feeding rate | $a(dose) = \frac{a(control)}{a(dose)}$ | $a = \frac{1}{e^{\beta_{p,control}+\beta_{p,dose}}} + \frac{1}{e^{\beta_{f,control}+\beta_{f,dose}}}$ | $\beta_{p,control}$ = 8.03 (7.88, 8.17) $\beta_{p,low}$ = 7.97(7.78, -8.16) $\beta_{p,high}$ = 7.70(7.53, 7.88) $\beta_{fcontrol}$ = 7.00 (6.92, 7.06) $\beta_{f,low}$ = 7.37(7.26, 7.49) $\beta_{f,high}$ = 7.55(7.43, 7.67) | 0.78 (0.69, 0.88) |
| | Fraction of baseline oviposition rate | $o(dose) = \frac{a_f(control)}{a_f(dose)}$ | $a_f = \frac{1}{e^{\beta_{f,control}+\beta_{f,dose}}}$ | | 0.69 (0.59, 0.78) |
| Lethality | Probability of first day mortality | $\mu(x_{dose})$ | $e^{\beta_0+\beta_1 x_{dose}}$ | $\beta_0$ = -4.34 (-5.66, -3.32) $\beta_1$ = 2.15(1.39, 3.10) | 0.11 (0.07, 0.16) |
| | Fraction time until death relative to baseline | $\phi(dose)$ | See S3 Table | See S3 Table | 0.75 (0.61, 0.91) |
| Entry and exit rates | Probability of being repelled | $\rho(dose)$ | Markov Chain model as defined in [22] | $\rho$ | 0.19 (0.03, 0.36) |
| | Fraction of baseline exit rate | $q_T/q_U$ | Markov Chain model as defined in [22] | $q_T/q_U$ | 0.71 (0.40, 1.06) |

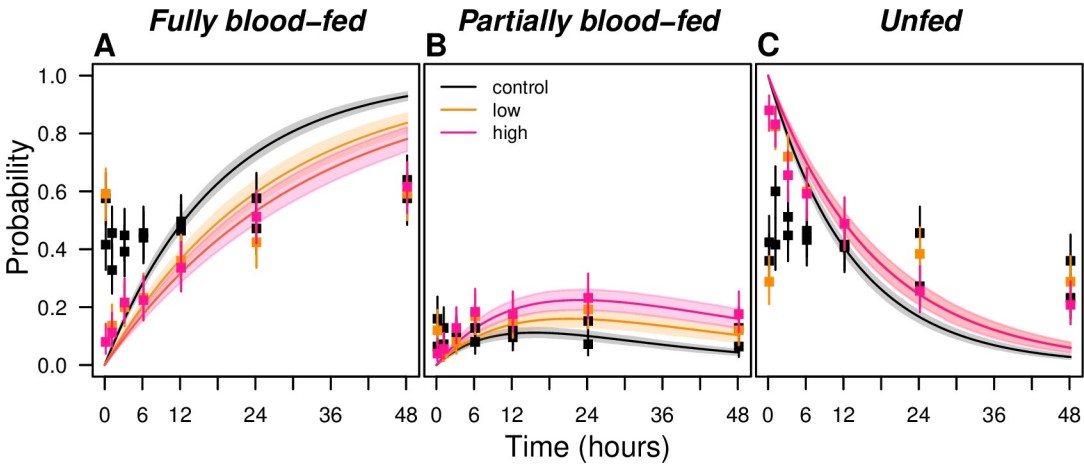

**Fig 2. Effect estimates of an experimental SR product containing transfluthrin on the probability of blood feeding over time.**
(A) fully blood-fed, (B) partially blood-fed, and (C) unfed *Aedes aegypti* mosquitoes. Squares denote the observed data for control (black), low (orange), and high dosage regimen (pink). High and low dosage estimates in C overlap. Bars denote the corresponding binomial 95%-confidence intervals. The shaded areas denote the 2.5th and 97.5th percentile of the model estimates

feeding were similar (S1 Table, see S3 Table for model description). Further, when expanding the model to include instantaneous feeding and a probability of never feeding, we find, based on AIC, that these models better fit the data (S2 Table, S2 Fig). Specifically, mosquitoes exposed to the control product were more likely to feed directly (62%) than when exposed to a low (42%) or high (20%) dosage regimen. Mosquitoes under control or low scenarios are estimated to feed eventually, whereas 39% of high exposed mosquitoes are expected to never feed whatsoever. When comparing the average time at which 50% of mosquitoes have fed, the general impact of the SR is relatively robust to the different model assumptions. Treatment effects are estimated to be somewhat larger under the more complex models (S2 Table).

**Exposure to transfluthrin resulted in both acute and delayed mortality.** We performed experiments to estimate the effect of exposure to low ($8.4 \times 10^{-7}$ g/L) and high ($12.6 \times 10^{-7}$ g/L) dosages of transfluthrin on mosquito mortality. We distinguished two product effects: 1) acute lethality ($\mu$), which occurs within a relatively short timeframe after exposure, and 2) delayed lethality ($\phi$), where lethal product effects have a prolonged effect on mosquitoes and eventually result in shorter life spans compared to unexposed mosquitoes.

We fitted a random binomial model to the number of deaths that occurred during the first day of the lethality experiments and showed that there was a significant effect of transfluthrin, and its dosage, on the probability of mortality shortly after exposure to the SR ($\mu$) ($P = 2.8 \times 10^{-14}$, t-test, df = 74) of 11% (HDI: 7–16%) (low) and 33% (HDI: 23–44%) (high) (Table 2).

We performed a survival analysis on mortality beyond day one and estimated that deaths that occurred after the first day showed a significant effect of transfluthrin, and its dosage, on what we regarded in our model as delayed lethality ($\phi$) (Table 2), with mean time until death reduced to 75% (HDI: 61–91%) at low dosage and 53% (HDI: 42–65%) at high dosage. Although these results are based on an assumption of exponentially distributed time until death, estimates of $\phi$ based on alternative parametric models with different assumptions about time-varying hazards of mortality were similar (S2 Table).

**Exposure to transfluthrin resulted in reduced entry to and exit out of treated spaces.** We performed experimental hut studies to measure the effect of exposure to low ($0.0025$ g/m$^2$) and high ($0.005$ g/m$^2$) dosages of transfluthrin on rates of entrance into treated spaces (repellency) and exit out of treated spaces (expellency). The results of this study have been published

earlier [22]. Here, we briefly present those parts of the results that are of relevance for the presented model framework.

We fitted a continuous-time Markov chain model to the trap data and estimated that repellency (ρ) was highest at a low dosage of transfluthrin, with a median proportion of 19% (HDI: 3–36%) of mosquitoes being deterred by the product and opting to move away from rather than towards the treated hut. Repellency was lower at the higher dosage of transfluthrin, with a median of 8% repellency (HDI: -9-25%) (S5A Fig). The mosquitoes that did enter the treated hut were estimated to have exited at a lower rate, suggesting disorienting hyperactivity responses of transfluthrin in the experimental hut study. At low dosage, median exit rates ($q_T$) were a factor 0.70 (HDI: 0.41–1.09) relative to exit rates from an untreated hut ($q_U$). A similar effect was estimated at high dosage (0.69, HDI: 0.40–1.06).

## Model-based projections of community-level epidemiological impacts

**Projections of community-level impacts of a SR product indicated substantial reductions in the force of infection, largely driven by lethality.**   For an SR product with characteristics similar to those quantified in our experiments, the total estimated community-level impact on relative force of infection (FoI) was projected to be substantial, with a 50% reduction in FoI estimated at 24% coverage (HDI: 14–38%) at low dosage and a 50% reduction in FoI estimated at 8% coverage (HDI: 5–12%) at high dosage (Fig 3A). The nonlinear relationship of this effect indicates strong indirect effects of the SR when community-level effects are accounted for. A large portion of this effect was attributable to the lethality of the product (Figs 3A and 4). This is a result of the cubic scaling of reductions in mosquito lifespan on FoI; i.e., reducing mosquito density, reducing the probability that a mosquito becomes infected in its lifetime, and reducing the probability of surviving the incubation period and thus being able to transmit the pathogen [23].

**Projections of community-level impact of a SR product without lethality indicated modest reduction in force of infection due to increased probing.**   An SR without any lethality (i.e, μ = 0, $g_T = g_U$) was estimated to have a more modest impact on FoI (Fig 3B). When

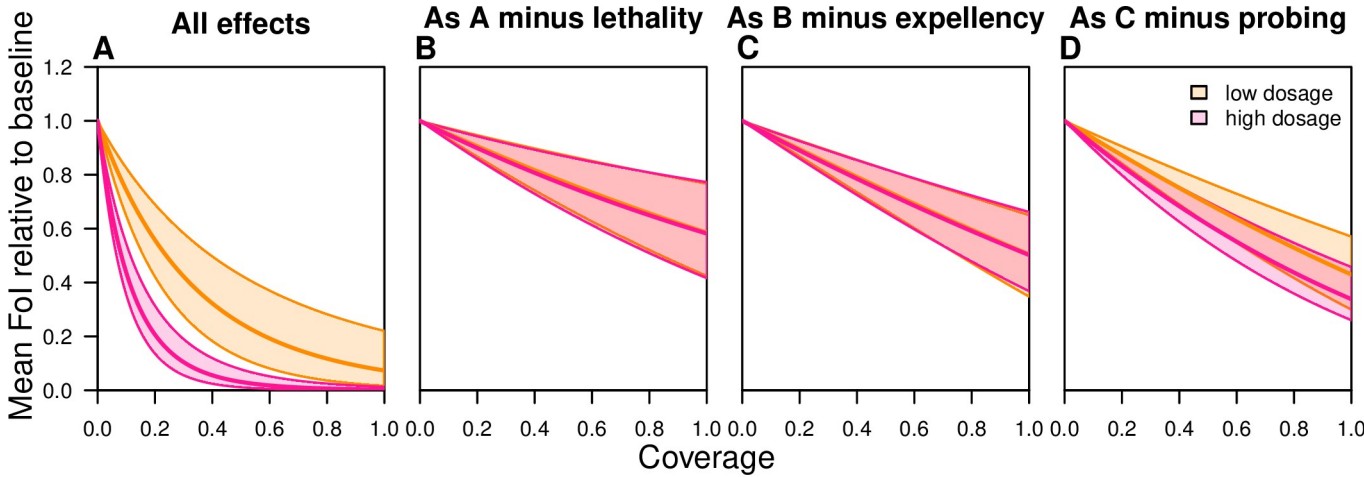

**Fig 3. Composite effects of an experimental SR product containing transfluthrin on relative force of infection (FoI) as a function of coverage across houses in a community.** (A) all effects, i.e, for this product formulation increased (partial) blood feeding or probing, delayed oviposition or time until fully blood fed, increased acute and delayed mortality, reduced entry rates (repellency), and reduced exit rates (expellency). (B) as A without lethality effects, (C) as B without expellency, and (D) as C without probing effect, with the median default estimates (solid lines) and the 2.5th and 97.5th percentile for the low (orange) and the high (pink) experimental dosage.

considering the combined effects on blood-feeding and repellency without reduced exiting, the maximum estimated reduction in FoI at low dosage was 50% (HDI: 35–63%) at 100% coverage, and similar for high dosage (50%, HDI: 35–62%) (Fig 3C). The increased propensity for partial blood feeding limits the community-level impacts of the SR in our model. In the absence of a partial blood-feeding effect (i.e., $o_C = a_C$), the maximum impact of this product would further increase at low dosage to a reduction in FoI of 57% (HDI: 44–71%) at 100% coverage and 66% (HDI: 55–77%) at high dosage (Fig 3D).

**The role of expellency and repellency on community-level impacts of a SR product depend on the context and product profile.** Exposure to transfluthrin in the hut experiments reduced the rate at which mosquitoes exited the huts. This reduction in the exit rate ($q_T/q_U = 0.70$) prolongs the time to take a blood meal indoors, thereby increasing the probability of transmission at that location. This relative enhancement of FoI (Fig 4Ff) is greater if outdoor mortality is higher relative to indoor mortality (S6J Fig) because the prolonged time spent indoors reduces exposure to outdoor hazards. Conversely, if a SR has strong effects on delayed mortality, reduced exit rates can increase exposure to the product, resulting in substantial reduction in FoI (Fig 4Df). If a SR with a positive effect on exiting the treated space (e.g., $q_T/q_U = 1.30$) were to be used, the result of more bouts transiting between houses would enhance the maximum community-level impact of the SR (in the absence of lethality) from a 41% (HDI: 23–60%) reduction in FoI at low dosage to a 56% (HDI: 43–69%) reduction at similar indoor and outdoor death rates ($g_U/g_\tau = 1$). If outdoor mortality were three times higher than indoor mortality, this would enhance the maximum population-level impact of the SR to an 81% (HDI: 69–92%) reduction in FoI. Similar reasoning holds for the impact of repellency effects.

## Sensitivity analysis

We found that the estimated effects of the SR were robust to some but not all free parameters. The results were sensitive to the extrinsic incubation period (EIP, n), the baseline mortality rate ($g_U$), the ratio between the indoor and outdoor mortality rate ($g_U/g_\tau$), and the ratio between average time indoors versus outdoors ($q_U/q_\tau$) (S6A, S6G, S6H and S6J Fig). We found that an SR product with characteristics similar to those in our model were less effective if the EIP (n) were longer and baseline mortality rates ($g_U$) were higher. The risk of not surviving the EIP—and not successfully transmitting DENV—was increased under such assumptions, resulting in relatively smaller impacts on FoI of SR-induced mortality. When mortality was higher outdoors than indoors, the effect of the SR was also reduced under the scenario of decreased exiting from the treated space. This was due to a lower proportion of time spent outdoors than if the treatment were absent. This pattern was reversed when a product that increases exit rates or elicits stronger repellency was assumed. Lastly, the effect of an SR product with characteristics similar to those in our model was reduced in the event that mosquitoes spend more time in transit to another house relative to their residence time indoors (assuming $g_U = g_\tau$). In such a setting, mosquitoes encountered the SR less often over the course of their lifetimes, thereby diminishing the effect of the SR.

## Discussion

We have introduced a model for making projections of community-level impacts of spatial repellent (SR) products on diseases vectored by *Ae. aegypti*. The impacts captured by our model derive not only from individual protection but from a combination of direct and indirect effects at different levels of product coverage within a community. Using a suite of laboratory and experimental-hut experiments, we parameterized six distinct effects of an

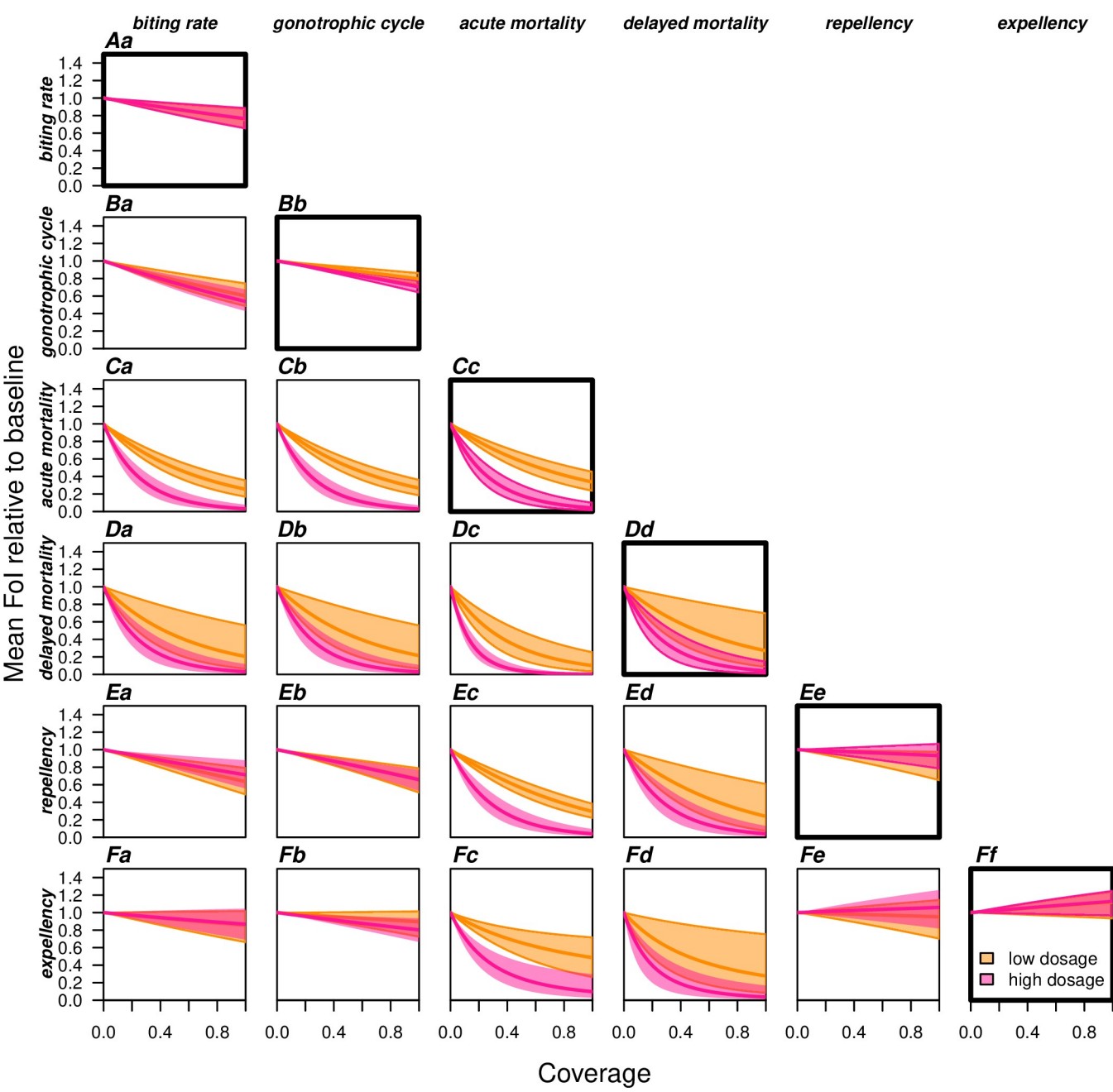

**Fig 4. Composite effects of an experimental SR product containing transfluthrin on relative force of infection (FoI).** Effects were assessed as a function of population coverage for different modes of action (A.-F.) (Table 2). The diagonal (bold, Aa-Ff) shows each product effect by itself. Off-diagonal planes show each product effect combined with one other product effect (.a-.f). The mean default estimates (solid lines) and the 2.5th and 97.5th percentile are shown for the low (orange) and the high (pink) experimental dosage.

experimental transfluthrin-based SR product on mosquito behavioral and bionomic traits. This example product was not an optimized, commercially formulated product, but rather served to exemplify the suite of effects a single product may elicit and how these effects can act in concert to reduce transmission. We demonstrated that a product with a profile similar to this one could have a substantial epidemiological impact by way of reducing the force of infection (FoI) of a pathogen transmitted by *Ae. aegypti* in a community with appreciable product

coverage. This is largely driven by the product's lethality, but not completely. For instance, when deployed in a mosquito population that has gained partial resistance to lethality [24] a product such as the one in our analysis could still lead to a meaningful reduction in FoI due to multiple forms of delayed blood feeding.

One potential use of our model is as a tool to guide the design of new products and to assess their potential impact when deployed at scale. As a demonstration of this capability, we found that the transfluthrin treatment examined in this study significantly increased mortality and reduced overall biting and hut entry. On the other hand, the product was found to result in increased partial blood feeding and delayed exiting from treated spaces. The relative probability of transmission following partial and full blood feeding is unclear, but increased partial blood feeding could negatively affect the epidemiological impact of the product if the probability of transmission is similar to what it is with full blood feeding [25]. Conversely, an increase in number of feeds could be accompanied with increased mortality due to the risks involved with blood feeding, which would further enhance the impact of the SR [26–28]. How such hazards differ in response to product exposure remains to be investigated. Irritancy, inducing effects such as increased partial blood feeding and repellency, was common at low, sub-lethal dosages [13] and could affect the potential net benefit of a SR product. This finding is consistent with earlier work showing that the modes of action of active chemicals depend on the dosage of the product, with irritancy being more prevalent at lower dosages [13, 22]. This underscores the importance of understanding how dosage affects multiple behavioral and bionomic traits. How these effects attenuate as active chemicals decay over space and time will also be critical to quantify, to enable better prediction of downstream effects on neighboring premises and to assess the product life and associated replacement strategies. Field experiments that allow for measurement of multiple effects in conjunction and across space and time would increase accuracy of effect estimates by allowing environmental and exposure conditions to be matched [22].

The projected epidemiological impact of SRs and other vector control products depends on both the characteristics of the products (active chemicals, application format) and the context of their deployment. We found the potential impact of the example SR product in this study to be highest in densely populated urban areas where mosquitoes are assumed to spend a relatively large portion of time indoors and transit relatively quickly between houses [14]. In these settings, mosquitoes may be more likely to have frequent encounters with the SR and thereby to be affected by the lethal, irritant, and excitatory effects. However, if the SR has stronger repellency and expellency effects (*i.e.*, increased exit rates), such as the metofluthrin formulation used in [6], longer transit times in sparsely populated areas would result in longer biting delays and increased transit-related mortality, resulting in a further reduction of transmission. Notably, high-dosage SRs could have reduced impacts in very densely populated areas at low coverage. In that situation, the repellency effect could prevent mosquitoes from entering a house with high SR-associated mortality. A delay in biting as a result of the SR would then be offset by the reduced life span it would have experienced if it had entered the house. At low coverage, densely populated settings could be most prone to the adverse effects of diversion of infected mosquitoes to untreated homes. The potential for diversion is strongly tied to the risks a mosquito experiences from its exposure to the product and from the hazards a mosquito experiences when moving from one house to the other. In settings where transit is relatively hazardous, SRs that reduce time indoors are expected to have a greater impact. Push-pull strategies, which trap mosquitoes in sentinel traps after they have been repelled or expelled from a house [29], may therefore be a promising candidate in this context.

Field studies are needed to validate the presented framework and disentangle the bionomics most affected by interventions in real-world settings. From an entomological point of view,

these evaluations should integrate the deployment of entry and exit traps [30–32] and measurements of mosquito abundance in treated and untreated places [8]. Mark-release-recapture studies in (semi-)field settings could further contribute to estimating the potential for mosquito diversion and the circumstances under which this may or may not be a concern. Further, studies aiming at estimating how bionomic effects impact FoI require combined entomological and epidemiological efforts. Large-scale, cluster-randomized trials that examine the epidemiological impacts of community-level roll out of SRs, on seroconversion and/or disease incidence, should be appended with entomological indices as described above [7, 8]. The presented framework is intended to be used to analyze such population-level studies and help pinpoint the bionomic effects that have the biggest impact on FoI or those that present barriers for impactful implementation.

Capturing realistic effects of product exposure in laboratory experiments is not straightforward. Exposure in the used experiments was short, yet continuous, resulting in a strong effect shortly after exposure. In the field, longer term, intermittent exposures are more likely and are expected to illicit a more gradual effect on mosquito-bionomics. The latter would more closely resemble model assumptions that rates and product effects are on average constant over time. For this framework, the main interest lies in relative effects of product exposure. Estimated relative effects were robust to different assumptions on time-varying hazards, justifying our simplifying assumptions. In other words, while more detailed models could provide a better fit to the experimental data, these would in part capture experimental artifacts that are not *per se* reflective of a natural response to product exposure. An additional challenge deriving from the experimental data used, is the potential for bias. Both blood feeding effects and delayed mortality are measured on a population of mosquitoes that did not die of acute lethality effects. It is possible that these individuals are less susceptible to the measured effects than the overall mosquito population, leading to underestimations of the product effects. Realism of model predictions could be improved by experimental set-ups that i) more closely resemble natural exposure and ii) measure different product effects in parallel. Extending the experimental hut study [22] with metrics of blood feeding could facilitate this.

The Ross-McDonald framework used has a number of limitations that may affect the outcome of SR implementation. Bites are modeled to be homogeneously distributed, assuming individuals to be equally attractive to mosquitoes and mosquitoes distributing their bites equally over time and across houses. Departures from homogeneous mixing can affect predictions of transmission dynamics and the impact of control in both directions (such as explored in [33–36]). Typically, homogeneous mixing results in more efficient transmission, because bites are well distributed across individuals. More detailed model frameworks are beneficial for disentangling the impact of transmission heterogeneity. These could also be used to cross-validate more parsimonious and transparent frameworks, such as the one presented here. While differences in transmission are expected under more heterogeneous transmission scenarios, earlier studies comparing agent-based models with Ross-McDonald-like frameworks showed the latter to be adequate in revealing the relative synergistic effects of vector control (purpose of present study) and can do so in full transparency [37].

Further, space is not explicitly modeled. For instance, our model assumes that each house has an equal probability of being visited by a mosquito, irrespective of its proximity to the house where the mosquito was previously. In addition, the model assumes that no biting occurs in transit. This does not, however, exclude the possibility of outdoor biting. The houses in the model should be regarded as the totality of all space affected by the SR and may thus include semi-enclosed and open areas in sufficiently close proximity of the product. Further, we assumed that placement of the product was random. A clustered rollout may be logistically desirable and could, depending on the context and the formulation of the product, result in

community-level impacts that differ from our estimates. One possible advantage of clustered rollout could occur if the product has beneficial downstream effects to adjacent, untreated houses. Such effects are currently not included in our framework, but they could enhance the impact of the product unless adverse, irritating effects such as partial blood feeding or reduced expellency occur downstream.

Downstream effects (*i.e.*, in adjacent, untreated spaces) of the experimental transfluthrin treatment were observed in the experimental hut study (25). These include reduced expellency (for both dosages) and lethality (only for the high dosage) effects. Downstream effects on blood-feeding rates are currently unknown but may well occur in tandem with other irritating effects such as reduced expellency [6]. In addition, dosage decay over time may alter the product profile by increasing irritating effects and decreasing effects on mortality as the product expires. Although our framework does not allow for temporal changes in treatment effects, it does allow for the exploration of a suite of different profiles. Given appropriate experimental data on effects following the decay of the product, our framework could be used to estimate the cross-sectional effects of a SR at different stages of the product's lifetime as a way to inform best practices for replacement timing. Coupling more detailed models with additional laboratory and hut experiments would aid in capturing the full effects of heterogeneity in exposure and protection, both on treated and untreated spaces. The model presented here provides a general way to gain insights on the projected interplay of different behavioral and bionomic effects of a SR in a variety of settings.

The sustained burden of dengue and the emergence of other pathogens transmitted by *Ae. aegypti* mosquitoes highlight the need for novel vector control paradigms [2]. Spatial repellents are one promising tool for settings where other vector control products are insufficient, such as the control of day- and outdoor-biting mosquitoes, in settings with high potential for resistance against lethality, or as an additional tool in outbreak response. Our model could likewise be extended to other (combinations of) vector control tools with multifaceted effects on mosquito behavioral and bionomic traits, including push-pull regimens and window screening, similar to how alternative modeling frameworks for malaria have been used to address effects of multiple interventions [15, 16]. In general, our model offers a way of synthesizing the results of feasible experiments at small scales to meet public health challenges at large scales.

## Materials and methods

Our study involved 1) the development of a new model to project the epidemiological impact of vector control products that elicit multifaceted effects on *Ae. aegypti* behavioral and bionomic traits; 2) the collection and analysis of experimental data to quantify those effects; and 3) sensitivity analyses of the model-based estimates of community-level impact informed by experimental results. The experiments comprising the second component of our study all involved transfluthrin but did so with differing means of application or delivery of that chemical. As application methods may result in different levels of volatile particles that mosquitoes are exposed to, we cannot directly compare the relative product impacts on mosquito bionomics. Consequently, the third component of our study should be regarded as a hypothetical, but empirically grounded, analysis rather than a specific analysis of a singular product formulation.

### Model

To estimate the community-level impact of vector control products that potentially affect transmission through multiple behavioral and bionomic traits, we built on existing frameworks for modeling mosquito-borne pathogen transmission [38]. We assumed a well-mixed

community meaning that each house has an equal probability of being encountered by a mosquito. Each house has a probability $C$ of being protected by the product at any given time. Consequently, upon searching for a blood meal, mosquitoes encounter a treated house with probability $C$, assuming the possibility of repellency of the product can only occur once the mosquito has "encountered" the house (*i.e.*, chemical cue) at close proximity. Below, we describe the effects that the product may have on a mosquito once it makes such an encounter, how those effects scale with coverage $C$, and how those scaled effects impact the force of infection of a mosquito-borne pathogen transmitted in a community in which these interventions are deployed (Fig 1).

**Entomological effects–blood-feeding.**   We assume that blood-seeking mosquitoes encounter a SR product with probability $C$, upon which they are repelled with probability $\rho$. After this occurs, mosquitoes move at rate $1/\tau$ to another house. Due to various excitatory/agitation behavioral responses associated with the product, a mosquito that does enter a treated house feeds at a rate that is a proportion $\alpha$ of the mean blood feeding rate, which has an average length $1/a$ in the absence of the SR. In addition, the expellency caused by the intervention alters the time a mosquito spends in the house from $1/q_U$ to $1/q_T$, on average.

To arrive at an estimate of the overall delay in blood feeding that results from the combination of these effects, we first calculated the average delay associated with each of these events weighted by the probability of each such event. In the event that a mosquito encounters a treated house but does not enter it, which occurs with probability $C\rho$, the delay before the next house is visited is simply the transit time $\tau$. In an untreated house, which mosquitoes enter with probability 1-$C$, a mosquito fails to blood-feed before leaving the house with probability $\frac{a}{a+q_u}$. The average delay associated with this is $q_U^{-1}+\tau$. In a treated house, which mosquitoes enter with probability $C(1-\rho)$, a mosquito fails to blood-feed before leaving the house with probability $\frac{\alpha a}{\alpha a+q_T}$. The average delay associated with this is $q_T^{-1}+\tau$. Together, the probability of one of these three events occurring is $D = C\rho + C(1-\rho)\frac{\alpha a}{\alpha a+q_T} + (1-C)\frac{a}{a+q_u}$, and the expected delay conditional on one of these events occurring i

$$\delta = \frac{\tau C\rho + (q_T^{-1} + \tau)\left(C(1-\rho)\frac{\alpha a}{\alpha a+q_T}\right) + (q_U^{-1} + \tau)((1-C)\frac{a}{a+q_u})}{D}.$$

(1)

We next considered that multiple such delays could occur consecutively. If experiencing one of these delays from one blood-feeding attempt to the next are independent events, then the expected number of delays before successful blood feeding is $D / (1 - D)$. To capture these effects within a single modified biting rate $a_C$, we can equate the reciprocal of that biting rate with the average time until a successful blood meal, which is

$$\frac{1}{a_C} = \delta\frac{D}{1-D} + \frac{1}{a}.$$

(2)

**Entomological effects–mortality.**   In the event that a mosquito is undeterred by the repellent effect of the product, it becomes exposed to the lethal effects of the product and is assumed to die with probability $\mu$ within a relatively short timeframe after entering the house. In addition, exposure to the product may have delayed lethal effects that increase the baseline death rate of the mosquito ($g_T$). Lastly, a mosquito that is successfully repelled by the product may experience mortality later—e.g., due to predation—while it is transitioning to another house, experiencing a mortality rate $g_\tau$ during transit [39, 40]. The interplay of these lethal effects augments the background mortality rate $g$ in the absence of vector control to result in a new overall mortality rate $g_C$ in the presence of a SR product at coverage $C$. The new overall mortality

rate follows from the sum of rates associated with each of three states (in transit: $\Delta$; present in a treated house: T; present in an untreated house: U), weighted by the proportion of time spent in each state.

To derive the proportion of time spent in each state, we use a continuous-time Markov chain model of the whereabouts of a mosquito in one of three states $\{\Delta, U, T\}$ over time and use that model to derive an expectation for the long-term average of time allocation across those three states. That model is driven by the infinitesimal matrix [41]

$$\mathbf{A} = \begin{bmatrix} \tau^{-1}(\rho C - 1) & \tau^{-1}(1 - C) & \tau^{-1}(1 - \rho)C \\ q_U & -q_U & 0 \\ q_T & 0 & -q_T \end{bmatrix}, \tag{3}$$

where the rates in each element of the matrix determine the instantaneous probability of a mosquito moving from one state to another and are consistent with parameter definitions and assumptions in the previous section on delayed blood-feeding.

The transition probabilities $P_{ij}(t)$ satisfy a system of differential equations with rates $\mathbf{A}$ known as the backward Kolmogorov differential equations [41]

$$\frac{dP}{dt} = \mathbf{A}\mathbf{P}(t), \tag{4}$$

where $\frac{dP}{dt}$ follows from the rates of change in each state

$$\frac{d\Delta}{dt} = \tau^{-1}(\rho C - 1)\Delta + q_U U + q_T T$$

$$\frac{dU}{dt} - q_U U + \tau^{-1}(1 - C)\Delta \tag{5}$$

$$\frac{dT}{dt} = -q_T T + \tau^{-1}(1 - \rho)C\Delta.$$

At equilibrium, $\frac{dP}{dt} = 0$ and so the stationary distribution $\lim_{t \to \infty} \mathbf{P}(t) = \pi = (\pi_\tau, \pi_U, \pi_T)$ follows from eqn. (4) and the notion that the probabilities must sum to one. Solving $0 = \mathbf{A}\mathbf{P}(t)$ under that condition gives the probabilities

$$\pi_\tau = -\frac{\tau q_U q_T}{q_U C(\rho - 1) + q_T(C - \tau q_U - 1)}$$

$$\pi_U = \frac{(1 - C)\pi_\tau}{\tau q_U} = \frac{(C - 1)q_T}{q_T(C - \tau q_U - 1) + C(\rho - 1)q_U} \tag{6}$$

$$\pi_T = \frac{(1 - \rho)C\pi_\tau}{\tau q_T} = \frac{C(\rho - 1)q_T}{q_T(C - \tau q_U - 1) + C(\rho - 1)q_U}$$

that a mosquito is in a given state at any given time. The average mortality rate $g_C$ under coverage $C$ then follows by taking the probabilities from eqn.(6) and using them to weight the state-specific probabilities $\mu$, $g_T$, and $g_U$ according to

$$g_C = \pi_T(\mu q_T + g_T) + \pi_U g_U + \pi_\tau g_\tau. \tag{7}$$

Here, instantaneous lethality $\mu$ is divided by the average time spent in a treated house ($q_T^{-1}$) to approximate the rate per day at which this effect acts on an average individual present in a treated house. Further, by weighing state-specific death rates by the time spent in a specific state, we implicitly assume that delayed mortality effects of treatment are proportional to exposure time. Due to the Markovian nature of the model (i.e., hazards a mosquito experiences in

its current states are independent of past states), mortality effects do not carry over to other states.

**Entomological effects–mosquito density.** We assume that effects of the SR product on mosquito density act through effects on demographic processes; *i.e.*, birth and death. Assuming abundant breeding sites and thus limited impact of density dependent processes, a general form for equilibrium mosquito density is $m = \varepsilon / g$, where $\varepsilon$ is the rate of emergence of new adult mosquitoes and $g$ is death rate. One formulation of the emergence rate $\varepsilon$ is that it is the product of the expected number of blood meals that each mosquito takes over the course of its lifetime and a combination of immature-stage mortality and development rates [38]. We treat the latter as an unspecified constant and the former as the product of the rate of oviposition $o$ and the expected lifetime $1/g$, implying that $m_C \propto o_C/g_C^2$. While both $o_C$ and $g_C$ are affected by the time spent indoors and outdoors, we make no assumptions on where oviposition takes place. Oviposition therefore has no effect on where mosquitoes spend their time.

**Epidemiological impact.** We use the force of infection (FoI) as our focal metric for quantifying the epidemiological impact of a SR product. FoI is defined as the rate at which susceptible individuals become infected. In Ross-Macdonald models of mosquito-borne pathogen (MBP) transmission, FoI = $bmaY$, where $b$ is the probability that a human becomes infected after being bitten by an infectious mosquito, $m$ is the ratio of mosquitoes to humans, $a$ is the rate at which a mosquito engages in blood feeding (either partial or full blood feeding), and $Y$ is the prevalence of infection among mosquitoes [23]. The latter depends further on the daily mosquito mortality rate $g$, the incubation period $n$ in the mosquito, the probability $c$ that a mosquito becomes infected upon biting an infectious human, and the prevalence of infection in humans $X$ according to

$$\text{FoI} = bmaY = \frac{bma^2cXe^{-gn}}{g + acX}. \tag{8}$$

Under equilibrium assumptions in an SIS-SI malaria model, $X$ can in turn be solved for as a function of model parameters [38]. Under non-equilibrium assumptions, however, the formulation in eqn. (8) is equally valid for other types of compartmental models (e.g., SEIR-SEI) provided that $X$ is regarded as a free parameter, which is appropriate given its highly dynamic and uncertain nature for strongly immunizing MBPs such as DENV.

To derive an expectation for how FoI will change in response to a SR, we can take the ratio of the expression in eqn. (8) evaluated with parameter values reflecting the intervention at a given coverage $C$ against the expression evaluated with parameter values reflecting conditions in the absence of the SR. We allow for the possibility that any or all of $m$, $a$, $o$, and $g$ may change in response to the SR. This results in a ratio of FoIs of

$$\text{FoI}_{rel} = \frac{\text{FoI}_C}{\text{FoI}} = \frac{m_C a_C^2}{ma^2} \frac{g + acX}{g_C + a_C cX} e^{-(g_C - g)n}. \tag{9}$$

Substituting the result $m_C \propto o_C/g_C^2$ into eqn. (9), we obtain

$$\text{FoI}_{rel} = \frac{o_C a_C^2 g^2}{oa^2 g_C^2} \frac{g + acX}{g_C + a_C cX} e^{-(g_C - g)n}, \tag{10}$$

which effectively depends on the effects of the intervention on rates of blood feeding and mortality, as well as three free parameters, $C$, $X$, and $n$. Uncertainty around fitted parameters is compounded to obtain the uncertainty around FoI$_{rel}$. Specifically, we calculated FoI$_{rel}$ 10,000 times each time taking an independent random draw from the multivariate normal

distribution encompassing all estimated parameters. The 95% highest density interval (HDI) is then derived by finding the smallest interval that contains 95% of the estimates for FoI$_{rel}$.

## Collection of experimental data

**Mosquito populations.**   Laboratory experiments to measure effects on blood feeding and mortality were performed using colonized, pyrethroid-resistant adult *Ae. aegypti* ($F_5$) from the Department of Entomology, Katsetsart University, Thailand. Field experiments to measure effects of repellency and expellency were performed using $F_{1-4}$, pyrethroid-susceptible adult *Ae. aegypti* maintained from field-collected immatures in Iquitos, Peru (100% susceptible, CDC Bottle Bioassays, exposure time 2 hours, diagnostic time after 30 minutes).

**SR effect on blood feeding.**   *Ae. aegypti* of age 5–7 days ($N = 200$ for each arm), starved for 24 hrs pre-test, were exposed to transfluthrin or solvent alone (control) for 10 minutes using a high-throughput screening mechanism following previously described protocols [42]. An hour after exposure, mosquitoes that were not knocked down due to exposure were selected ($N = 125$ for each arm) and allowed to feed on human blood (Interstate Blood Bank, Inc, Type O CPDA-1 Whole Blood) using a membrane feeding system. After a predefined follow up time, mosquitoes were dissected and recorded as fully blood-fed, partially blood-fed, or unfed. The experiment was repeated for seven follow-up times (0, 1, 3, 6, 12, 24, and 48 hours) for each of three different dosages (control, low: $8.4 \times 10^{-7}$ g/L, and high: 1.5 times low dosage). Each experiment was performed at 80˚F and 75% humidity (Data are deposited at: DOI 10.17605/OSF.IO/J9CKS).

**SR lethality.**   *Ae. aegypti* of age 5–7 days ($N = 120$ for each arm) were exposed to transfluthrin or control as described above, after which cohorts were maintained at 80˚F and 75% humidity with access to 10% sugar solution. Adult survival was monitored daily for 25 days. The experiment was repeated for five different treatment dosages relative to the $8.4 \times 10^{-7}$ g/L (0: control, 0.5, 0.75, 1, 1.25, 1.5). Dosages 1 and 1.5 are hereafter referred to as low and high dosage, respectively (Data are deposited at: DOI 10.17605/OSF.IO/J9CKS).

**SR effect on entry and exit rates.**   To quantify rate of entry and exit of *Ae. aegypti* exposed to transfluthrin under field conditions, an experimental hut study was performed in Iquitos, Peru [22]. A unique hut design was employed in which five experimental structures were aligned in a single row with adjoining walls containing open eave gaps to create a continuum of indoor space. This design mimicked a housing structure common to Iquitos, Peru, as well as other dengue-endemic areas. Each 4m x 6m x 2m hut was equipped with exit traps. The center hut contained cotton treated with transfluthrin or solvent alone (control, technical grade aceton) during baseline experiments. In each of the huts, a human was present under an untreated bed net to generate host-seeking cues and monitor mosquito knockdown (inability to fly). For each experiment, 5–7 day old, sugar-fed mosquitoes ($n = 25$, $F_{2-3}$), uniquely marked according to individual huts, were released inside all but the center hut at 5:30 AM. Exit traps were checked every 30 minutes from 6:00 AM until 6:00 PM, and knockdown was monitored hourly within this same time frame. At 6:00 PM, remaining mosquitoes were recaptured from inside each hut using aspirators, and their release and recapture locations were recorded based on marking color scheme. The experiment was repeated five times for three different treatments (control, low: 0.0025g/m$^2$, high: 0.005g/m$^2$) (Data are deposited at: DOI 10.17605/OSF.IO/5HCPF).

## Analysis of experimental data

**SR effect on blood feeding.**   We used data collected in the blood-feeding experiments to estimate partial ($a_p$) and full blood feeding ($a_f$) rates as a function of the dose of transfluthrin.

Details of this analysis are described in Supporting Information. Briefly, we assumed that blood feeding follows a Poisson process with dose-dependent rates $a_p(dose) = e^{-(\beta_{p,0}+\beta_{p,dose})}$ and $a_f(dose) = e^{-(\beta_{f,0}+\beta_{f,dose})}$. For each of the three possible outcomes (partially blood-fed, fully blood-fed, unfed) the probability of observing the outcome was derived, while taking note that the number of fully blood-fed mosquitoes observed in the experiment includes both mosquitoes that were fully engorged after one meal and those that became fully engorged after multiple partial blood meals (Supporting Information). Using these probabilities, we maximized the multinomial likelihood of the model parameters given the observed number of mosquitoes in each feeding category using the *bbmle* package [43] in R [44] and in doing so obtained, for each dosage, best-fit estimates of the coefficients $\beta_{p,0}$, $\beta_{p,dose}$, $\beta_{f,0}$, and $\beta_{f,dose}$ describing the effects of the SR on the biting rates $a_p$ and $a_f$.

**SR lethality.** We used data collected during the lethality experiments to estimate two distinct lethal product effects on acute ($\mu$) and delayed ($\phi$) lethality. To separate these effects, we consider observed death and survival over the course of day one to be informative of $\mu$ and any death and survival thereafter to be informative of $\phi$.

Over the course of day one, we assumed that the number of mosquito deaths was a binomial random variable with probability $\mu$ with relationship to transfluthrin dose $x_{dose}$ defined by

$$\mu(x_{dose}) = e^{-\beta_0 + \beta_1 x_{dose}}. \tag{11}$$

We obtained maximum-likelihood estimates of $\beta_0$ and $\beta_1$ using the *bbmle* package [43] in R [44]. While the first time-point was after 24 hrs, we assume that mosquitoes that die during the first day either die sooner or are sufficiently affected to no longer contribute to relevant epidemiological processes (*i.e.*, blood feeding and reproduction) and thus can reasonably be assumed 'removed from the population' shortly after exposure.

To estimate the delayed lethality effect of the SR (*i.e.*, conditional on survival beyond day one), we fitted different parametric survival models to the right-censored time-to-event data collected in the laboratory experiments [45]. The methods for this analysis are detailed in Supporting Information. Briefly, we sought to estimate the effect of chemical dosage on the hazard over time. The hazard $\lambda$ is the instantaneous rate at which death occurs and is described as the probability that death occurs between time $t$ and time $t+dt$, given that $dt$ is small and death has not occurred before $t$. The specific form of the hazard model that we sought to estimate was a function of the transfluthrin dose, expressed at time $t$ as

$$\lambda_i(t|dose) = \lambda_0(t)e^{\beta_1,dose}, \tag{12}$$

where $\lambda_0(t)$ is the baseline hazard for individuals with *dose* equal to zero (*i.e.*, the control group) and $e^{\beta_1,dose}$ is the relative hazard associated with a specific dosage, which was assumed to be stable over time (*i.e.*, proportional hazards) (see S4 Fig and Supporting Information for further details and validation of this assumption). Under the assumption that the distribution of survival times results from a continuous-time stochastic process, we used parametric survival models to describe $\lambda_0(t)$ [45], each with different assumptions on how $\lambda_0(t)$ changes over time (S4 Fig). We used the Akaike Information Criterion [46] to select the best model while penalizing for more parameters in more complex models. The models were fitted and assessed using the *survival* package 2.27–7 [47] and the *flexsurv* package [48] in R [44].

**SR effect on entry and exit rates.** Data from the hut experiments included competing interval-censored data on exit (30-minute intervals) and knockdown (hourly intervals) by event hut and release hut. In addition, mosquitoes recaptured at the end of the experiment represent right-censored data. Mosquitoes not recaptured at any point during the experiment

were considered to have been lost to follow-up at some unknown time before the end of the experiment.

In brief, ten Bosch et al. [22] used a continuous-time Markov Chain (CTMC) model to describe the probability at a given time for a mosquito, given its release location, to be present in a given hut or to have experienced a given outcome. At any time, a mosquito can be in one of five huts (transient states: $H_{2L}$, $H_{1L}$, $H_0$, $H_{1R}$, or $H_{2R}$), or have already experienced one of 15 terminal events: exit, knockdown, or loss to follow-up in any of the five huts (absorbing states: $X_i$, $K_i$, and $U_i$ $\forall i \epsilon [2L, 1L, 1R, 2R]$). The rates at which mosquitoes transition either to adjacent huts or to an absorbing state are assumed to be independent of time or previous trajectories, and thus the time spent in each transient state is exponentially distributed. The parameters of the CTMC model were fitted to the data using a Bayesian Markov chain Monte Carlo (MCMC) approach coded in the R language [44] and using the *coda* package [49] for processing of the results.

## Modeling analysis

**Baseline model parameterization.**    Six distinct entomological effects of the transfluthrin treatment evaluated are defined by the probability of death upon encountering the SR ($\mu$), increases in mortality rates ($\phi$), the probability of repellency ($\rho$), the rate of change in exit rates ($q_T/q_U$), delayed blood feeding ($\alpha$) expressed as a proportion of the mean rate of blood feeding in the absence of SR, and delayed oviposition ($o$) (Table 2). Estimates of these parameters were derived from the lethality, blood-feeding, and entry and exit rate experiments described above.

The probability of death shortly after entering a treated hut ($\mu$) follows directly from eqn. (11). Note that we assume this value to be zero in untreated spaces and thus solely a result of the product. Short-term mortality observed in the control experiment is assumed to be captured in the baseline mortality rate ($g_U$). From the second part of the analysis of the mortality experiment, we estimated the reduction in time until death in response to the SR relative to the control group ($\phi$) from the accelerated failure time survival models. The treatment-adjusted mortality rate is

$$g_T = g_U/\phi. \tag{13}$$

From the blood-feeding experiment, we estimated the blood feeding rate ($a_T$) in a treated house as the fraction $\alpha$ of this quantity relative to the biting rate ($a_U$) in an untreated house, where we assume that the ratio of biting rates in treated and untreated houses is the same as the ratio of the biting rates in the treatment and control arms of the laboratory blood-feeding experiment. The oviposition rate ($o$) is estimated similarly, but solely relies on the rate at which mosquitoes become fully blood fed.

Repellency ($\rho$) is informed by the proportion of mosquitoes leaving an adjacent hut ($H_{1L}$ or $_{1R}$) that move away from rather than towards the treated hut. Assuming that in an untreated environment mosquitoes move to either neighboring hut with an equal probability (*i.e.*, $p_1 = 0.5$), the repellency effect relative to the untreated scenario is $\rho = 1 - (1 - p_{1T}/0.5)$. Here, we assume that the relative effect of $\rho$ is similar for mosquitoes being repelled from outside as from entering from a neighboring space, even though the overall entry rate might be different between these two scenarios. The rate at which a mosquito exits a treated house relative to an untreated house follows from the fitted exit rates in [22]. The relative exit rate out of the treated house is a fraction $q_T/q_U$ of the default rate $q_U$ (Table 1).

**Sensitivity and scenario analysis.**    For the baseline parameterization of the model, we assumed that exposures across experiments map to similar dosages in the population setting. This assumption is necessary for illustrative purposes, but it should be kept in mind that the

level of exposure in real-life settings will differ from any experimental setting, in particular for the laboratory experiments. We examined a range of different combinations of bionomic effects to assess the sensitivity of our estimates to different product profiles and to quantify uncertainty associated with our estimates. Further, we examined the impact of different assumptions on the free parameters (Table 1). We investigated the sensitivity of our relative FoI estimates across a range of plausible values. In addition, the population-level effects of SRs may vary across settings. For instance, whereas densely populated areas could result in shorter travel times between houses ($\tau$), this duration and associated hazards could be elevated in less densely populated settings due to increased distance and more abundant predators. The impacts of different transit times ($\tau$), transit hazards ($g_\tau$), and residence times ($1/q$) on our estimates of relative FoI by coverage were assessed in a similar fashion as the other free parameters.

## Supporting information

**S1 Text. Materials and methods**
(DOCX)

**S1 Fig. Estimated dose effects of an experimental SR product containing transfluthrin on mosquito blood feeding.** (A) exponential, (B) Weibull, (C) lognormal, and (D) gamma models. The dashed lines depict the Kaplan-Meier curves at associated dosages.
(DOCX)

**S2 Fig. Effect estimates of an experimental SR product containing transfluthrin on the probability of blood feeding over time for different model alternatives.** (A,D,G) fully blood-fed, (B,E,H) partially blood-fed, and (C,F,I) unfed *Aedes aegypti* mosquitoes under control (A,B,C), low (D,E,F), and high (G,H,I) dosage regimens. We distinguish a model that i) assumes constant biting rates (solid), ii) as i but preceded by a proportion of mosquitoes feeding directly at the start of the experiment (dashed), and iii) as ii but with a proportion of mosquitoes that will never feed altogether (dotted). Squares denote the observed data for control (black), low (orange), and high dosage regimen (pink). Bars denote the corresponding binomial 95%-confidence intervals.
(DOCX)

**S3 Fig. Estimated dose effects of an experimental SR product containing transfluthrin on mosquito longevity.** (A) exponential, (B) Weibull, (C) lognormal, (D) gamma, and (E) generalized gamma models. The dashed lines depict the Kaplan-Meier curves at associated dosages (control, low, high).
(DOCX)

**S4 Fig. Proportional hazard test for longevity data conditioned on first day survival.** The proportional hazards assumption holds for dosage regimen that are parallel to each other when plotted with these transformations.
(DOCX)

**S5 Fig. Posterior estimates of effects of an experimental SR product containing transfluthrin.** (A) repellency (decreased entry) and (B) expellency (increased exit) during exposure to control, low (0.0025 g/m2), and high (0.005g/m2) dose regimen.
(DOCX)

**S6 Fig. Sensitivity of relative force of infection (FoI) estimates to the baseline parameters as a function of population coverage.** (A) extrinsic incubation period from 3 to 33 days, (B)

transmission probability from mosquito to human from 0.01 to 1, (C) as B but from human to mosquito, (D) human infection prevalence from 1 to 99%, (E) the duration of the gonotrophic cycle from 1 to 14 days, (F) the baseline biting rate from 0.2 to 10, (G) the baseline mosquito mortality rate from 0.025 to 2.5, (H) mortality rate during transit relative to indoor mortality rate from 0.1 to 10, (I) average time spent in an untreated house from 0.05 to 5 days, and (J) the proportion of time a transit event takes relative to the baseline residence time, from 0.1 to 10. Across all panels, yellow depicts low values and dark red signifies high values.
(DOCX)

**S7 Fig. Posterior distributions of model parameters fitted to experimental data for the baseline (gray), low (orange) and high (pink) transfluthrin dosage for the treated hut (subscript T) and huts one or two removed from the treated hut (subscript 1 and 2, respectively). (reproduction from Figure 6 in [22]).** Posterior distributions of model parameters fitted to experimental data for the baseline (gray), low (orange) and high (pink) transfluthrin dosage for the treated hut (subscript T) and huts one or two removed from the treated hut (subscript 1 and 2, respectively). (a-c) Rates at which mosquitoes exit the huts. (d) Proportion of movement from H1 (hut directly adjacent to the treatment hut) away from the SR, where the dashed line indicates p1 = 0.5, i.e. no repellency effect. (e-g) Knockdown rates. (h) Loss to follow-up rates. Under this parameterization, the movement rate $q_i$ is exactly equal to the product $x_i / r_i$. The algorithm was run for 90,000 iterations inclusive of a 'burn-in' period of 10,000.
(DOCX)

**S1 Table. Comparing functional forms of transfluthrin effects on blood feeding (low dosage, $8.4\text{x}10^{-7}$ g/L).**
(DOCX)

**S2 Table. Comparing models on transfluthrin effects on time until 50% of mosquitoes blood fed (low dosage, $8.4\text{x}10^{-7}$ g/L).** The AIC (Akaike Information Criterium) denotes the model fit, with a lower fit presenting a better fit [46].
(DOCX)

**S3 Table. Transfluthrin effects on mortality by survival model (low dosage, $8.4\text{x}10^{-7}$ g/L).**
(DOCX)

**S4 Table. Survival functions.**
(DOCX)

## Acknowledgments

We thank Theeraphap Chareonviriyaphap for materials and support for experiments conducted in Thailand and Amy Morrison, Roxanne Burus, and Victor Lopez for materials and support for experiments conducted in Peru. We thank Angela Carranci and Suppaluck Polsomboon for assistance with experimental data collection. We thank Edwin Requena, Hugo Jaba and the field team for their support during the experiments in Peru. We thank Neil Lobo, Tom Scott, David Smith, and Steve Stoddard for useful discussions about spatial repellents as vector control tools. We thank Dyon van Essel and Infograaf for artwork.

The views expressed in this work are those of the authors and do not reflect the official policy or position of the Department of the Navy, Department of Defense, or U.S. Government.

## Copyright statement

FCL is employed of the U.S. Government. This work was prepared as part of her official duties (PJTNMRCD 0.24). Title 17 U.S.C. § 105 provides that 'Copyright protection under this title is

not available for any work of the United States Government'. Title 17 U.S.C. § 101 defines a U. S. Government work as a work prepared by a military service member or employee of the U.S. Government as part of that person's official duties.

## Author Contributions

**Conceptualization:** Quirine A. ten Bosch, Joseph M. Wagman, Fanny Castro-Llanos, Nicole L. Achee, John P. Grieco, T. Alex Perkins.

**Data curation:** Joseph M. Wagman, Fanny Castro-Llanos.

**Formal analysis:** Quirine A. ten Bosch.

**Funding acquisition:** Nicole L. Achee.

**Investigation:** Quirine A. ten Bosch, Joseph M. Wagman, Fanny Castro-Llanos, John P. Grieco, T. Alex Perkins.

**Methodology:** Quirine A. ten Bosch, T. Alex Perkins.

**Validation:** Nicole L. Achee.

**Writing – original draft:** Quirine A. ten Bosch, Nicole L. Achee, T. Alex Perkins.

**Writing – review & editing:** Joseph M. Wagman, Fanny Castro-Llanos, John P. Grieco.

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
