## [Decision Letter · Decision Letter 0]

13 Nov 2019

Dear Dr ten Bosch,

Thank you very much for submitting your manuscript 'Community-level impacts of spatial repellents for control of diseases vectored by Aedes aegypti mosquitoes' for review by PLOS Computational Biology. Your manuscript has been fully evaluated by the PLOS Computational Biology editorial team and in this case also by independent peer reviewers. The reviewers appreciated the attention to an important problem, but raised some substantial concerns about the manuscript as it currently stands. While your manuscript cannot be accepted in its present form, we are willing to consider a revised version in which the issues raised by the reviewers have been adequately addressed. We cannot, of course, promise publication at that time.

Sincerely,

Jennifer A. Flegg

Guest Editor

PLOS Computational Biology

Virginia Pitzer

Deputy Editor

PLOS Computational Biology

[LINK]

Reviewer's Responses to Questions

**Comments to the Authors:**

Reviewer #1: The authors present a compelling framework to assess a broad range of public-health impact and product-development profiles through the interpretation of targeted small-scale entomological experiments combined with a rational model framework.

It is especially valuable that this paper takes a purposefully broad view of potential combinations of distinct spatial-repellent effects and maps out the relevance of different components and combinations of components on population-level effects -- including a thoughtful sensitivity analysis.

-----------

The only major concern is the handling of the fits in Figure 2, particularly the mismatch between the chosen functional forms and what is realized in the control experiments. This is mentioned in the text (L161-166) and explored to some degree in S1 Table, but unlike for survival -- which has a graphical exploration (S1 Fig) and quantitative assessment of fit quality (S3 Table) -- the sensitivity of downstream results to interpretation of blood-fed data remains a bit unclear.

At a minimum, another supplemental figure demonstrating the fit quality of the scenarios in S1 Table in relation to Figure 2 would be appreciated.

It might also be a useful activity to explore scenarios where some fraction of experimental mosquitoes (~30%) will never feed, where not all partial feeds are "non-absorbing" state (~10%), or where mosquito feeding is a bi-modal phenomenon (first ~40% feed within hours, next ~20% feed over days).

-----------

Minor comments:

* There introduction of model parameters and the mathematical symbols feels a bit informal in some places. For example, referring to "o=Relative gonotrophic cycle length" in Figure 1, "delayed oviposition" on L134, and "oviposition rate" on L160.

* A more accurate formatting of highest density intervals might also include a % symbol after the ranges (where appropriate).

* The horizontal axis of Figure 2 would be easier to interpret if the tick marks and labels were in hours, e.g. 6h, 12h, 18h.

* Figure 4: If it is intended that related off-diagonal columns can be compared by eye between the two dosages, this is a challenge with the current design choice to use the above and below diagonal halves in this way.

* L282 and S4 Fig: Where do the univariate sensitivity ranges come from and how much do those choices impact the apparently most sensitive variables?

* L338: "chemical actives"  "active chemicals"?

* L369: "per see"  "per se" (italic)

* L541: Is this a robust assumption, or does the relationship of adult densities to oviposition rates have some uncertainty in larval density-dependent dynamics?

* The companion paper (ref #25 in Parasites & Vectors) has a reference now, it seems, so this submission should be updated accordingly. If journal restrictions do not prevent it, the current manuscript would certainly benefit from some reproduction of Figure 6 in the supplement.

Reviewer #2: The manuscript presents a mathematical model of the impact of spatial repellents on the force of infection of dengue. The model is based on laboratory and semi-field research with Aedes aegypti. Model parameters included direct mortality, but also sublethal effects of spatial repellents on host-seeking, blood feeding, and flight. I have the following comments.

General comments. There is no question that the manuscript represents a very large research effort. The manuscript, however, is very tedious to read and the writing is often unclear (see specific comments below). Additionally, I have some concerns about the methodology. The Abstract (lines 21-22) and Figure 1 indicate that the model is based on the premise that mosquito production occurs outdoors forcing host-seeking females to enter houses to acquire a blood meal. However, in the semi-field experiments conducted in Iquitos, Peru, mosquitoes were released within houses. Authors please respond to this concern.

Laboratory experiments were conducted with a pyrethroid “resistant” strain of Aedes aegypti, but field experiments were carried out using a “susceptible” strain. No data on the toxicity of transfluthrin or other pyrethroid insecticides to either strain is presented to support these characterizations. I am concerned that the response of resistant and susceptible mosquitoes to transfluthrin would be different so that some parameters would be over/under estimated resulting in an erroneous model. Authors please comment on this concern.

Additionally, I am concerned about the use of an artificial membrane system to evaluate host-seeking and blood-feeding by SR-exposed mosquitoes. It has been my experience that mosquitoes are less responsive to an in vitro system than to a human forearm. If the lab strain was verified to be pathogen-free, why weren’t the mosquitoes allowed to feed on a human host?

Specific comments.

1. Abstract, lines 26-27. “…chemical’s lethal effect but delayed biting and associated negative feedbacks on the mosquito population…” Please clarify what is meant by “associated negative feedbacks”.

2. Lines 29-31. Please be specific about what is meant by “potential adverse impacts”.

3. Authors’ summary. Lines 43-46. “…negative feedbacks on the mosquito population, elicit its own substantial impact.” Again, the meaning of this statement is not clear.

4. Lines 46-47. “Adverse effects of increased partial blood-feeding and reduced exiting could offset gains achieved by other effects.“ Please be specific about the “other effects”.

5. Introduction, lines 54-57. “…more effective..." You just stated that there were setbacks in development of effective vaccines, suggesting that vaccines were not effective.

6. Lines 72-75. “…can promote adverse behaviors…” Why would these behaviors be adverse if the end result is successful blood feeding and oviposition?

7. Lines 116-119. “…distinguishing probing from time until oviposition.” These are two different behaviors and it is not clear how they are connected. Do the authors mean “blood feeding” rather than “probing”?

8. Line 120. “…the product effect is hereby referred to as ‘lethality’…” So the myriad effects that the authors attribute to spatial repellents are included in one parameter “lethality”?

9. Line 121. “…hazards due to transit…” Authors please consider rephrasing this. The manuscript involves entomology not engineering.

10. Results. Lines 150-153. “This result was largely driven by a reduction in the rate at which full blood meals were taken, with the average time until a full blood meal increasing relative to the control by 46% (HDI: 27-65) (low) and 74% (HDI: 50-100) (high).” Does the reduction also result from delays in successful host-seeking?

11. Line 154. “…partial blood meals (probing effect)…” Are the authors suggesting that partial blood meal result from aberrant probing?

12. Lines 156-157. “…increased insignificantly relative to the control in response to low exposure by 5% (HDI: -19-26) but did increase after high exposure 28%...” Was the 28% increase statistically significant?

13. Line 207. “Repellency was lower at the higher dosage of transfluthrin…” A counter intuitive finding that the authors should explain in the discussion section.

14. Line 283-284. “We found that an SR product with characteristics similar to those in our model were less effective if the EIP (n) were longer and baseline mortality rates (gU) were higher.” Don’t the sublethal effects of SRs increase the EIP? It is not clear why a high baseline mortality rate leads to a decline in the effectiveness of SR products.

15. Discussion, line 325. “…increased feeding-induced mortality…” Please explain what this means. Is there a supportive reference that the authors can cite?

16. Lines 330-332. “How these effects attenuate as chemical actives decay over space and time…” Is this a potential weakness of an SR for prevention of mosquito bites and reducing the FOI? Unless an SR is continuously released then effects of the SR would be short lived because newly emerged adults would replace those impacted by the SR. Authors please comment on this.

17. Line 349. “…repellency effect could protect mosquitoes from entering a house…” Do the authors mean “…could prevent mosquitoes from entering a house…”?

18. Lines 353-355. ‘The potential for diversion is strongly tied to the risks a mosquito experiences both from its exposure to the product and from its transit between houses.” How can “transit between houses” be a cause of diversion?

19. Lines 366-370. “Estimated relative effects were robust to different assumptions on time-varying hazards, justifying our simplifying assumptions. In other words, while more detailed models could provide a better fit to the experimental data, these would in part capture experimental artifacts that are not per see [sic] reflective of a natural response to product exposure.” These sentences are poorly constructed and not informative. What artifacts are the authors referring to?

20. Materials and Methods. Lines 435-439. The authors are hedging on the validity of their model. They should speculate on how the difference in application methods would have affected their model.

21. Line 446. What is a “well-mixed community”?

22. Lines 447-450. This sentence is poorly constructed. “…assuming the possibility of repellency or attraction of the product…” Mosquitoes are attracted to an SR?

23. Lines 491-493. “…a mosquito that is successfully repelled by the product may experience additional mortality later—e.g., due to predation—…” Use of “additional” suggests that “the mosquito” died previously. Can the authors cite a published study on predation as a significant mortality factor for Aedes aegypti?

24. Lines 582-587. My concern about the use of resistant and susceptible mosquito strains is expressed above.

25. Lines 592-595. My concern about the use of an artificial feeding system is expressed above.

26. Lines 616-618. “…contained transfluthrin-treated material or solvent alone…” Please describe the material used to release transfluthrin and the solvent.

27. Line 626. “…three different treatments (control, low: 0.0025g/m2, high: 0.005g/m2)…” How were these release rates determined?

28. Line 627. “(DOI 10.17605/OSF.IO/5HCPF)” What does this mean?

29. Figure 1. My comments about Figure 1 have been give above.

30. Figure 2A, C. Control data appear to be poorly fit to the regression lines. Some measure of fit of the data points to the regression lines (e.g., R-square) should be given. In Figure 2C there is no regression line for the low dosage of transfluthrin.

Reviewer #3: The authors present an interesting experimental approach to epidemic

spreading and its sensitiveness to spatial repellents. The

work is up to my knowledge original and results seem correct.

In a revised version I suggest the authors to consider the

following two issues:

1) How can the modelling issues discussed in this paper be

applied with real data?

2) Some recent works use agent-based models to investigate the impact

of anti-malaria factors in epidemy spreading.

In Journal of Theoretical Biology 484 110030 e.g. an agent model is

introduced for assessing the influence of drugs and gametocitemia periods

in the spreading scenario, with a calibration using real data. Some

discussion in this scope would strengthen the paper in my opinion.

**Have all data underlying the figures and results presented in the manuscript been provided?**

Reviewer #1: None

Reviewer #2: Yes

Reviewer #3: None

PLOS authors have the option to publish the peer review history of their article (what does this mean?). If published, this will include your full peer review and any attached files.

Reviewer #1: Yes: Edward Wenger

Reviewer #2: Yes: Charles S. Apperson

Reviewer #3: No

---

## [Decision Letter · Decision Letter 1]

20 May 2020

Dear Dr. ten Bosch,

Thank you very much for submitting your manuscript "Community-level impacts of spatial repellents for control of diseases vectored by Aedes aegypti mosquitoes" for consideration at PLOS Computational Biology. As with all papers reviewed by the journal, your manuscript was reviewed by members of the editorial board and by several independent reviewers. The reviewers appreciated the attention to an important topic. Based on the reviews, we are likely to accept this manuscript for publication, providing that you modify the manuscript according to the review recommendations.

I agree with both reviewers that this manuscript is improved. However, further discussion on the validation of the model should be added, including what sorts of (future collection of) data could be used to validate the model further. In addition, a discussion of the limitations of the model and its underlying assumptions should be included (e.g. in the discussion section).

Sincerely,

Jennifer A. Flegg

Associate Editor

PLOS Computational Biology

Virginia Pitzer

Deputy Editor

PLOS Computational Biology

[LINK]

I agree with both reviewers that this manuscript is improved. However, further discussion on the validation of the model should be added, including what sorts of (future collection of) data could be used to validate the model further. In addition, a discussion of the limitations of the model and its underlying assumptions should be included (e.g. in the discussion section).

Reviewer's Responses to Questions

**Comments to the Authors:**

Reviewer #2: The authors have suitably revised their manuscript. The manuscript describes a robust investigation. Hopefully, the authors' intent is to conduct an area-wide field study to validate their model.

Reviewer #3: This version is better than the previous one. Still, I do not think

previous remarks on validation with empirical data and the limitations

(e.g. neglection of heterogeneous effects) were properly addressed.

I suggest the author to improve their manuscript in this scope.

**Have all data underlying the figures and results presented in the manuscript been provided?**

Reviewer #2: Yes

Reviewer #3: Yes

PLOS authors have the option to publish the peer review history of their article (what does this mean?). If published, this will include your full peer review and any attached files.

Reviewer #2: No

Reviewer #3: No
---

## [Editor Report · Decision Letter 2]

24 Jul 2020

Dear Dr. ten Bosch,

We are pleased to inform you that your manuscript 'Community-level impacts of spatial repellents for control of diseases vectored by Aedes aegypti mosquitoes' has been provisionally accepted for publication in PLOS Computational Biology.

Best regards,

Jennifer A. Flegg

Associate Editor

PLOS Computational Biology

Virginia Pitzer

Deputy Editor

PLOS Computational Biology

---

## [Editor Report · Acceptance letter]

22 Sep 2020

PCOMPBIOL-D-19-01743R2 

Community-level impacts of spatial repellents for control of diseases vectored by *Aedes aegypti* mosquitoes

Dear Dr ten Bosch,

I am pleased to inform you that your manuscript has been formally accepted for publication in PLOS Computational Biology. Your manuscript is now with our production department and you will be notified of the publication date in due course.

With kind regards,

Laura Mallard
